# Efficient Multi-View 3D Representation via Fusion of View-Agnostic Transformations

## Abstract

Bird's-Eye View representations are essential for 3D perception in autonomous driving, providing unified and spatially coherent scene understanding. While attention-based methods achieve strong performance through global cross-view attention, they suffer from computational inefficiencies due to redundant referencing and spatial ambiguity from ego-centric projections. To address these limitations, we introduce Mosaic View Transformation (MosaicVT), a modular framework that independently transforms multi-camera views into a unified BEV space. MosaicVT employs a camera-centric polar coordinate system, effectively resolving directional ambiguity and reducing cross-view redundancy. A novel view-agnostic positional embedding enables a single transformation module to generalize across heterogeneous camera configurations without retraining. Transformed camera-centric representations are then aligned and fused into a global BEV using a geometry-aware interpolation strategy, significantly reducing computational overhead compared to global attention mechanisms. Experimental results on the nuScenes benchmark demonstrate that MosaicVT achieves state-of-the-art performance in 3D object detection and BEV semantic segmentation while providing substantial reductions in latency and maintaining robust generalization across diverse camera setups.

## 1 Introduction

Bird's-Eye View (BEV) representation has emerged as a powerful framework for vision-based 3D perception, particularly in autonomous driving scenarios (Philion & Fidler, 2020; Li et al., 2024c; Liu et al., 2022). Unlike direct predictions made from individual camera views, BEV provides a unified and spatially coherent representation of the surrounding environment. This viewpoint aligns naturally with human intuition and facilitates effective fusion of sensor modalities such as LiDAR, enabling enhanced performance in downstream tasks such as 3D object detection, tracking, and semantic segmentation (Huang et al., 2021; Huang & Huang, 2022; Liu et al., 2023; Xie et al., 2022; Chen et al., 2022). Consequently, accurate and efficient construction of BEV representations from multi-view camera inputs has become a critical research direction for robust autonomous systems.

View Transformation (VT) is an essential step in the multi-camera BEV pipeline, responsible for projecting and aggregating multi-view image features into a unified BEV space. Explicit projection-based methods (Philion & Fidler, 2020; Xie et al.; Liu et al., 2023; Li et al., 2024b; Zhou et al., 2023) explicitly estimate per-pixel depth and then geometrically transform image features into BEV space. These approaches have demonstrated effectiveness through explicit geometric reasoning and accurate spatial grounding. Recently, complementary methods (Li et al., 2024c; Jiang et al., 2023; Yang et al., 2024) based on learnable transformations have been introduced. Instead of explicit geometric projection, these models leverage attention mechanisms to implicitly learn the spatial alignment between camera image features and BEV representation. This approach offers flexibility by learning spatial correspondences from data, without requiring precise geometric modeling.

Despite their promising results, existing transformer-based methods typically utilize global cross-view attention across image features from all cameras, which leads to computational inefficiency due to redundant referencing. As shown in Figure 1(a), attention patterns reveal that each BEV region predominantly attends to image areas closely aligned with its viewing direction, indicating that global attention across all views is largely unnecessary.

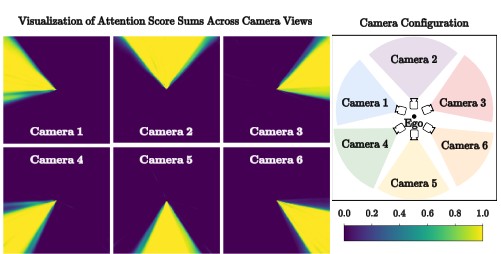 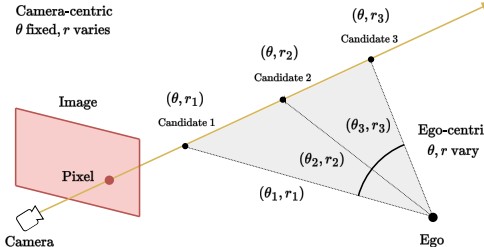

(a) Redundant attention to out-of-view cameras      (b) Ambiguity of pixel rays in ego-centric frame

Figure 1: **Redundant and ambiguous view transformation.** (a) Attention maps from a Transformer-based view transformation (Yang et al., 2024), showing per-camera sums of attention weights from each BEV query to image representations. Most weights concentrate on the camera observing the queried region, indicating that referencing other views is redundant. (b) A pixel corresponds to a ray in 3D space containing multiple possible points. In ego-centric coordinates, these points differ in both direction and distance, leading to spatial ambiguity. In contrast, the camera-centric frame fixes the direction and varies only in distance, simplifying the view transformation.

In addition, Figure 1(b) illustrates the ambiguity arising from pixel projection rays in ego-centric coordinates. Each image pixel corresponds to a ray that spans multiple candidate 3D points varying in both direction and radial distance relative to the ego-frame, complicating spatial alignment and hindering accurate feature integration.

To address these issues, we propose Mosaic View Transformation (MosaicVT), a modular framework that performs view transformation independently for each camera, grounded in its own geometric context. Rather than performing global reasoning , we adopt a camera-centric polar coordinate system that preserves the directional semantics of projection rays and naturally segments the BEV space by field of view. This formulation reduces ambiguity and cross-view interference by allowing each camera to contribute information within its own coverage region.

A core challenge of this design is ensuring that a single transformation module can operate across heterogeneous camera setups. We address this with a view-agnostic transformation that encodes image pixels using relative geometric cues: angular direction $\theta$, radial distance $r$, and height $h$ with respect to the ground. This representation abstracts away camera-specific parameters and places all views in a common relational frame, allowing the model to reason about spatial structure independently of the exact sensor pose. As a result, the same transformation network can generalize across different camera poses without requiring explicit calibration or view-specific customization.

After generating camera-centric BEV representations, we align them to a unified global frame through a learnable transformation conditioned on each camera's pose and field of view. The final global BEV is constructed by sampling from all aligned views using geometry-aware interpolation, which softly aggregates features based on spatial proximity. This process avoids global attention, preserves spatial continuity across views, and enables efficient fusion without introducing redundancy.

This design leads to a representation that is both spatially consistent and computationally efficient. By avoiding global cross-view attention and restricting transformation to local, geometry-aware operations, MosaicVT significantly reduces computational overhead while maintaining high fidelity in feature alignment. Experimental results on the nuScenes (Caesar et al., 2020) benchmark confirm that our method achieves state-of-the-art performance in 3D object detection and BEV map segmentation. Compared to transformer-based baselines, MosaicVT offers superior accuracy with substantially lower latency, and maintains robust performance even under variations in camera configuration.

Our contribution is summarized as follows:

- We propose MosaicVT, a framework that performs view transformation independently in camera-centric coordinates, reducing cross-view redundancy and directional ambiguity.
- MosaicVT outperforms prior methods in accuracy while offering significantly lower latency, and remains robust across diverse and unseen camera configurations without retraining.

## 2 RELATED WORKS

### 2.1 BEV-BASED 3D PERCEPTION

BEV representation has become central to 3D perception by enabling spatially consistent fusion of multi-view images. Modern pipelines extract image features, transform them to a top-down BEV space, and apply task-specific heads.

**3D Object Detection**    In BEV-based 3D object detection, center-based anchor-free heads (e.g., CenterPoint (Yin et al., 2021)) are widely adopted in methods such as BEVDet (Huang et al., 2021) and BEVDepth (Li et al., 2023a), which apply convolutional detection heads directly over the BEV grid. Transformer-based detectors like BEVFormer (Li et al., 2024c) instead use learned BEV queries and deformable attention to aggregate image features. MosaicVT follows the convolutional detection paradigm and replaces only the view transformation module within the BEVDet framework.

**Semantic Segmentation**    For BEV-based semantic segmentation, dense pixel-level classification over the BEV grid is commonly used. MosaicVT adopts this strategy by plugging into the decoder of BEVFusion (Liu et al., 2023), modifying only the feature projection stage. Other approaches, such as PointBEV (Chambon et al., 2024), adopt sparse BEV representations for improved efficiency. Unified multi-task models like UniAD (Hu et al., 2023) and HDMapNet (Li et al., 2022) demonstrate the benefits of sharing BEV features across detection, segmentation, and planning.

MosaicVT targets only the view transformation stage and integrates seamlessly with existing detection and segmentation heads.

### 2.2 BEV VIEW TRANSFORMATION

View Transformation (VT) projects multi-view image features into a unified BEV space. As this step involves ill-posed depth estimation, numerous methods have been proposed to enhance accuracy and efficiency. These approaches fall into three categories:

**Explicit Projection-Based Methods**    These methods estimate per-pixel depth distributions and explicitly lift 2D features into 3D space before projection to the BEV grid. LSS (Philion & Fidler, 2020) introduced this formulation with probabilistic depth. Fast-BEV (Li et al., 2024b) and MatrixVT (Zhou et al., 2023) improve runtime via height-pooling or structured projection.

**Implicit Attention-Based Methods**    Rather than relying on explicit geometry, these models learn spatial alignment through attention. BEVFormer (Li et al., 2024c) applies deformable cross-attention to aggregate multi-camera features into BEV queries. WidthFormer (Yang et al., 2024) enhances this with width-wise pooling and constrained attention regions. However, such transformer-based models often suffer from redundant global referencing and high latency.

**Auxiliary Supervision and Hybrid Strategies**    To improve feature grounding, methods like BEVDepth (Li et al., 2023a) and SA-BEV (Zhang et al., 2023) incorporate depth or semantic supervision. Hybrid models such as FB-BEV (Li et al., 2023b) and DualBEV (Li et al., 2024a) combine explicit lifting with attention-based refinement. HeightFormer (Wu et al., 2024) models vertical structure before projection, while BEV-SAN (Chi et al., 2023) introduces vertical slice attention to improve 3D reasoning.

MosaicVT departs from these designs by applying single-view transformation independently for each camera, without relying on cross-view attention, auxiliary signals, or multi-view fusion.

## 3 MOSAIC VIEW TRANSFORMATION

We propose Mosaic View Transformation (MosaicVT), a unified and modular framework for constructing global BEV representations from multi-view camera inputs. Our goal is to transform multi-view images $\{I^v\}_v$ into a unified global BEV representation $F_{\text{BEV}}$ for 3D perception. To reduce cross-view redundancy and resolve localization ambiguity, we process each view independently in a camera-centric polar coordinate system. This naturally partitions the ego-plane by viewing direction and provides a consistent angular reference, enabling unambiguous 3D localization. Based on this formulation, we first extract an image feature map $F^v$ for each view $v$, then construct a polar BEV representation $F_{\text{cam}}^v$. These are transformed into globally aligned representations $F_{\text{BEV}}^v$, which are finally fused to produce the global BEV representation $F_{\text{BEV}}$.

**Camera Geometry on Ego-Plane**
To define a polar BEV representation for each camera $v$, we first compute its field-of-view (FoV) and orientation on the ego-plane. This requires estimating four geometric components: the ground-projected camera center $\mathbf{o}_{\text{ego}}^v$, the effective horizontal FoV angle $\text{FoV}_{\text{ego}}^v$, and the forward and rightward directions $\mathbf{z}_{\text{ego}}^v$ and $\mathbf{x}_{\text{ego}}^v$.

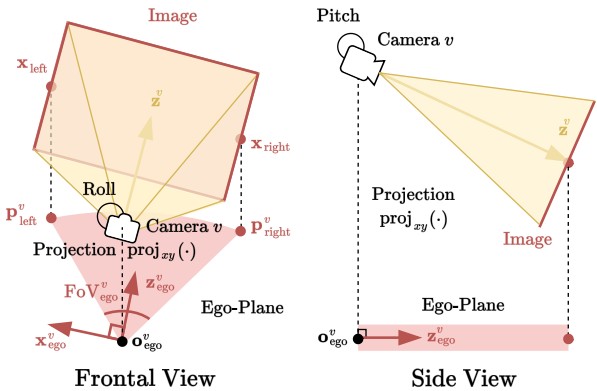

As shown in Figure 2, we define the effective field-of-view region by projecting representative image points onto the ground. Specifically, we select the midpoint pixels on the left and right image edges, $\mathbf{x}_{\text{left}}$ and $\mathbf{x}_{\text{right}}$, which indicate the left and right boundaries of the camera's FoV. These points are back-projected into 3D using the camera intrinsics $\mathbf{K}_v$, rotation $\mathbf{R}_v$, and translation $\mathbf{t}_v$ with the depth $\delta$ of the pixel in the camera coordinate system:

Figure 2: **Camera field-of-view region on ego-plane.** Frontal and side views showing how a camera's visible region is projected onto the ego-plane. To define the effective FoV region, we project rays from the midpoints of the left and right image edges. These midpoints are used instead of extreme edges to avoid angular regions covered by only a single pixel. The region is characterized by the ground-projected camera center $\mathbf{o}_{\text{ego}}^v$, its viewing direction $\mathbf{z}_{\text{ego}}^v$, lateral direction $\mathbf{x}_{\text{ego}}^v$, and the horizontal angular span $\text{FoV}_{\text{ego}}^v$.

$$\mathbf{p}_{\text{ego}}^v(\mathbf{x}, \delta) = \mathbf{R}_v(\delta \mathbf{K}_v^{-1}\mathbf{x}) + \mathbf{t}_v. \tag{1}$$

We then project the resulting 3D points onto the ego-plane using $\text{proj}_{xy}(\cdot)$:

$$\mathbf{p}_{\text{left}} = \text{proj}_{xy}(\mathbf{p}_{\text{ego}}^v(\mathbf{x}_{\text{left}}, \delta_0)), \quad \mathbf{p}_{\text{right}} = \text{proj}_{xy}(\mathbf{p}_{\text{ego}}^v(\mathbf{x}_{\text{right}}, \delta_0)). \tag{2}$$

Similarly, we define the ground-projected camera center as $\mathbf{o}_{\text{ego}}^v = \text{proj}_{xy}(\mathbf{t}_v)$. We compute vectors from the camera center to each boundary point, $\mathbf{p}_{\text{left}} - \mathbf{o}_{\text{ego}}^v$ and $\mathbf{p}_{\text{right}} - \mathbf{o}_{\text{ego}}^v$, and normalize them to obtain unit vectors $\mathbf{v}_{\text{left}}$ and $\mathbf{v}_{\text{right}}$. Given these vectors, the horizontal FoV angle is obtained by:

$$\text{FoV}_{\text{ego}}^v = \arccos(\mathbf{v}_{\text{left}} \cdot \mathbf{v}_{\text{right}}). \tag{3}$$

Finally, we define an orthonormal basis on the ego-plane:

$$\mathbf{z}_{\text{ego}}^v = \frac{\mathbf{v}_{\text{left}} + \mathbf{v}_{\text{right}}}{\|\mathbf{v}_{\text{left}} + \mathbf{v}_{\text{right}}\|}, \quad \mathbf{x}_{\text{ego}}^v = \frac{\mathbf{v}_{\text{right}} - \mathbf{v}_{\text{left}}}{\|\mathbf{v}_{\text{right}} - \mathbf{v}_{\text{left}}\|}. \tag{4}$$

### 3.1 VIEW-AGNOSTIC SINGLE-VIEW TRANSFORMATION

Rather than applying *separate transformations per view*, we propose a view-agnostic single-view transformation over each camera's region on the ego-plane. However, due to geometric differences between camera views, applying a shared transformation consistently is challenging.

Specifically, camera configurations determine the ego-plane region each camera covers and the 3D space each image pixel can represent. A pixel $\mathbf{x}$ in camera $v$ corresponds to a projection ray from the camera center, along which each 3D point can be described by its radial distance $r$ on the ground and height $h$. As illustrated in Figure 3, the shape of this $(r, h)$ distribution varies with the camera's vertical FoV, pitch, and mount height.

To address this challenge, we formulate view transformation as a *relative-coordinate-aware operation*, where each transformation is conditioned not on the image feature itself, but on its relative 3D location within the camera's region. This view-agnostic formulation enables a single transformation module to generalize across all cameras, despite their heterogeneous configurations.

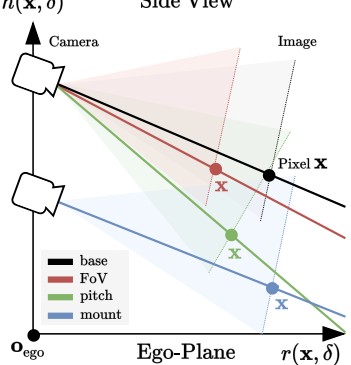

Figure 3: **Ray distribution shift.** Illustration of how variations in camera's vertical FoV, pitch, and mount height affect the $(r, h)$ distributions of rays toward the same image pixel $\mathbf{x}$, measured relative to the ego-plane.

Figure 4: **Mosaic view transformation (MosaicVT)**: (a) Each image is transformed into a camera-centric polar BEV representation, normalized by relative distance and FoV from the camera center for configuration invariance. (b) The representation is then aligned with the camera's FoV, orientation, and position on the ego-plane. (c) Transformed representations are spatially placed and fused into a unified global BEV map.

**View-Agnostic Positional Embedding** As a concrete realization of the relative-coordinate-aware formulation, we propose View-Agnostic Positional Embedding (VAPE), which represents each image feature in terms of its polar 3D location relative to the ego-plane: angular direction $\theta$, radial distance $r$, and height $h$ with respect to the associated camera view. Each component is normalized based on predefined geometric bounds to ensure consistency across views.

The angular coordinate $\theta$ is invariant to depth, as it depends solely on the projection direction of the pixel. It is computed at a fixed reference depth $\delta_0$ by transforming the pixel $\mathbf{x}$ into 3D, and measuring the angle between the direction vector and the local axes $\mathbf{z}_{\text{ego}}^v$ and $\mathbf{x}_{\text{ego}}^v$. The resulting angle is normalized by the horizontal field of view $\text{FoV}_{\text{ego}}^v$:

$$\hat{\theta}^v(\mathbf{x}) = \frac{2}{\text{FoV}_{\text{ego}}^v} \cdot \arctan\left(\frac{(\mathbf{p}_{\text{ego}}^v(\mathbf{x}, \delta_0) - \mathbf{o}_{\text{ego}}^v) \cdot \mathbf{x}_{\text{ego}}^v}{(\mathbf{p}_{\text{ego}}^v(\mathbf{x}, \delta_0) - \mathbf{o}_{\text{ego}}^v) \cdot \mathbf{z}_{\text{ego}}^v}\right) \in [-1, 1]. \tag{5}$$

In contrast, the radial distance $r$ and height $h$ vary with depth. For each depth $\delta$, we compute $r$ as the distance on the ego-plane from the ground-projected camera center $\mathbf{o}_{\text{ego}}^v$, normalized by a maximum range $r_{\max}$, and $h$ as the vertical coordinate of the 3D point, normalized by a height bound $h_{\max}$:

$$\hat{r}^v(\mathbf{x}, \delta) = \frac{\left\|\text{proj}_{xy}(\mathbf{p}_{\text{ego}}^v(\mathbf{x}, \delta)) - \mathbf{o}_{\text{ego}}^v\right\|}{r_{\max}}, \quad \hat{h}^v(\mathbf{x}, \delta) = \frac{[\mathbf{p}_{\text{ego}}^v(\mathbf{x}, \delta)]_z}{h_{\max}} \in [0, 1]. \tag{6}$$

To account for depth uncertainty, we predict a probability distribution $P_v(\delta|\mathbf{x})$ over depths and compute the expected values of $\hat{r}$ and $\hat{h}$ as:

$$\mathbb{E}_{\delta}[\hat{r}^v(\mathbf{x})] = \int \hat{r}^v(\mathbf{x}, \delta)\, P_v(\delta|\mathbf{x})\, \mathrm{d}\delta, \quad \mathbb{E}_{\delta}[\hat{h}^v(\mathbf{x})] = \int \hat{h}^v(\mathbf{x}, \delta)\, P_v(\delta|\mathbf{x})\, \mathrm{d}\delta. \tag{7}$$

In practice, the expectations are approximated using weighted sums over discrete depth bins. The resulting values $(\hat{\theta}^v, \mathbb{E}[\hat{r}^v], \mathbb{E}[\hat{h}^v])$ are embedded via sinusoidal encoding followed by a learnable MLP, producing the final VAPE embedding. This embedding provides a compact representation of each image feature's relative position in 3D space, consistent across views.

**Single-View Transformation** As shown in Figure 4(a), we compute the single-view polar BEV representation $F_{\text{cam}}^v$ using multi-head cross-attention:

$$F_{\text{cam}}^v = \text{MHA}_{\text{cross}}(Q_{\text{cam}}^v, F^v + \text{PE}_{\text{VAPE}}), \tag{8}$$

where each query $Q_{\text{cam}}^v(\hat{\theta}, \hat{r})$ encodes the relative angular and radial position of a BEV cell in the view's FoV. The query coordinates $(\hat{\theta}, \hat{r})$ are embedded via sinusoidal encoding followed by a learnable MLP, similar to VAPE.

The image features $F^v$ are augmented with VAPE embeddings of $(\hat{\theta}, \mathbb{E}[\hat{r}], \mathbb{E}[\hat{h}])$, which provide view-consistent 3D positional cues. Since $\hat{\theta}$ is deterministically computed from geometry, each image feature can be matched to a query with the same angular position, resolving directional ambiguity. The predicted radial distance $\mathbb{E}[\hat{r}]$ softly aligns with the query radial coordinate $\hat{r}$ through attention, and the paired height $\mathbb{E}[\hat{h}]$ encodes fine-grained spatial variation conditioned on camera pose.

## 3.2 BEV TRANSFORMATION

Once camera-centric polar BEV representations $F_{\text{cam}}^v$ are obtained, we transform them into the global BEV space for spatial consistency. Direct spatial warping is *insufficient*, as each polar BEV cell encodes directions and distances relative to its own camera. As shown in Figure 4(b), $F_{\text{cam}}^v$ alone lacks global consistency, motivating a transformation conditioned on scale, orientation, and translation. We therefore apply a *learnable representation-level transformation* $\mathcal{T}$ that incorporates three geometric factors:

1. *Scale*: $\text{FoV}_{\text{ego}}^v$ defines the extent of the polar grid.
2. *Rotation*: $\mathbf{z}_{\text{ego}}^v$ aligns the angular direction.
3. *Translation*: $\mathbf{o}_{\text{ego}}^v$ determines global placement.

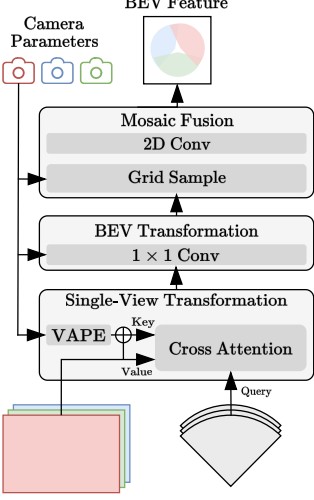

BEV Feature

Figure 5: **Overall architecture.** Detailed structure of MosaicVT.

As illustrated in Figure 5, we concatenate these cues with each camera BEV representation $F_{\text{cam}}^v$ and apply a shared $1 \times 1$ convolution:

$$F_{\text{BEV}}^v = \mathcal{T}(F_{\text{cam}}^v, \text{FoV}_{\text{ego}}^v, \mathbf{z}_{\text{ego}}^v, \mathbf{o}_{\text{ego}}^v). \tag{9}$$

This step produces a set of geometrically normalized BEV representations $\{F_{\text{BEV}}^v\}_v$, which are spatially aligned in a shared global coordinate frame and ready for fusion.

## 3.3 MOSAIC FUSION

The final global BEV representation $F_{\text{BEV}}$ is constructed by fusing transformed single-view BEV representations $\{F_{\text{BEV}}^v\}_v$ into a unified global space. Each transformed view-specific BEV $F_{\text{BEV}}^v$ contributes to the global representation based on its relative geometry, as illustrated in Figure 4(c). To ensure proper alignment, we determine the correspondence between each global BEV location $\mathbf{p}$ and the coordinates in each camera's local polar BEV grid. For each camera $v$, we compute a normalized polar coordinate $(\hat{\theta}_{\mathbf{p}}^v, \hat{r}_{\mathbf{p}}^v)$ as:

$$\hat{\theta}_{\mathbf{p}}^v = \frac{2 \cdot \arctan\left((\mathbf{p} - \mathbf{o}_{\text{ego}}^v) \cdot \mathbf{x}_{\text{ego}}^v / (\mathbf{p} - \mathbf{o}_{\text{ego}}^v) \cdot \mathbf{z}_{\text{ego}}^v\right)}{\text{FoV}_{\text{ego}}^v}, \quad \hat{r}_{\mathbf{p}}^v = 2 \cdot \frac{\|\mathbf{p} - \mathbf{o}_{\text{ego}}^v\|}{r_{\max}} - 1. \tag{10}$$

We then retrieve features from the polar BEV grid using a differentiable sampling function $\mathcal{S}$, which performs bilinear interpolation over neighboring grid points. Given a continuous sampling coordinate $\mathbf{q} = (\hat{\theta}, \hat{r})$, we define $\mathcal{N}(\mathbf{q})$ as the set of nearby discrete grid locations and $w(\mathbf{q}, \mathbf{q}')$ as the corresponding interpolation weights. The sampling function is then given by:

$$\mathcal{S}(F, \mathbf{q}) = \sum_{\mathbf{q}' \in \mathcal{N}(\mathbf{q})} w(\mathbf{q}, \mathbf{q}') F(\mathbf{q}'). \tag{11}$$

Let $\mathcal{V}(\mathbf{p})$ be the set of cameras whose effective fields of view spatially include $\mathbf{p}$. We average the sampled features across these views to obtain the final fused BEV representation:

$$F_{\text{BEV}}(\mathbf{p}) = \frac{1}{|\mathcal{V}(\mathbf{p})|} \sum_{v \in \mathcal{V}(\mathbf{p})} \mathcal{S}\left(F_{\text{BEV}}^v, (\hat{\theta}_{\mathbf{p}}^v, \hat{r}_{\mathbf{p}}^v)\right). \tag{12}$$

In implementation, this fusion process is realized by bilinear grid sampling followed by a 2D convolution, as illustrated in Figure 5. Since each view-specific BEV is transformed and interpolated placed into the global coordinate frame through geometry-aware interpolation, the overall procedure resembles composing a *mosaic* on the BEV plane. We refer to this unified and modular framework as Mosaic View Transformation (MosaicVT).

**Implementation Details** We use only a single-head cross-attention module over a polar BEV grid of resolution $16 \times 44$, and generate the global BEV representation at a resolution of $128 \times 128$. The model takes 6 camera images as input. Depth distributions for VAPE are predicted using a lightweight $1 \times 1$ convolution over discrete depth bins. The BEV transformation is implemented as a shared $1 \times 1$ convolution over concatenated geometric cues. Final fusion is performed via bilinear sampling followed by a $3 \times 3$ convolution.

Table 1: **3D object detection.** Comparison of BEV-based multi-camera view transformation methods on the nuScenes `val` set. The lower block shows results with stronger backbones and higher image resolutions.

| Method | Backbone | Resolution | mAP ↑ | NDS ↑ | mATE ↓ | mASE ↓ | mAOE ↓ | mAVE ↓ | mAAE ↓ |
|---|---|---|---|---|---|---|---|---|---|
| IPM (Kim & Kum, 2019) | Res-50 | 256×704 | 25.3 | 34.5 | 78.5 | 27.6 | 62.5 | 85.9 | 26.6 |
| LSS (Philion & Fidler, 2020) | Res-50 | 256×704 | 29.5 | 37.1 | 73.9 | **27.3** | 61.2 | 88.1 | 24.8 |
| MatrixVT (Zhou et al., 2023) | Res-50 | 256×704 | 28.9 | 36.5 | 74.6 | 28.3 | 60.0 | 89.5 | 27.3 |
| FastBEV (Li et al., 2024b) | Res-50 | 256×704 | 28.9 | 37.1 | 73.3 | 28.1 | 62.6 | **82.6** | 27.1 |
| BEVFormer (Li et al., 2024c) | Res-50 | 256×704 | 29.1 | 34.1 | 76.1 | 28.3 | 71.8 | 97.2 | 30.0 |
| WidthFormer (Yang et al., 2024) | Res-50 | 256×704 | 30.7 | 37.3 | 72.8 | 27.6 | 63.7 | 89.6 | 26.8 |
| **MosaicVT** | Res-50 | 256×704 | **31.6** | **39.3** | **68.9** | 28.2 | **54.7** | 89.4 | **24.4** |
| FCOS3D (Wang et al., 2021) | Res-101 | 900×1600 | 34.3 | 41.5 | - | - | - | - | - |
| DETR3D (Wang et al., 2022b) | Res-101 | 900×1600 | 34.6 | 42.5 | - | - | - | - | - |
| BEVDet (Huang et al., 2021) | Swin-B | 512×1408 | 34.9 | 41.7 | 63.7 | 26.9 | 49.0 | 91.4 | 26.8 |
| PGD (Wang et al., 2022a) | Res-101 | 900×1600 | 36.9 | 42.8 | - | - | - | - | - |
| PETR (Liu et al., 2022) | Res-101 | 512×1408 | 35.7 | 42.1 | 71.0 | 27.0 | 49.0 | 88.5 | **22.4** |
| BEVFormer-S (Li et al., 2024c) | Res-101 | 900×1600 | 37.5 | 44.8 | - | - | - | - | - |
| WidthFormer (Yang et al., 2024) | Res-101 | 512×1408 | 37.9 | 44.8 | 62.7 | **26.1** | 45.1 | **84.0** | 23.7 |
| **MosaicVT** | Res-101 | 512×1408 | **38.3** | **45.0** | **61.1** | 27.3 | 45.1 | 87.0 | **21.0** |

Table 2: **BEV map segmentation.** Comparison of BEV-based multi-camera models on the nuScenes `val` set. Metrics are per-class IoU scores (%) across semantic categories, with predictions made in the BEV plane.

| Method | Drivable ↑ | Ped. Cross. ↑ | Walkway ↑ | Stop Line ↑ | Carpark ↑ | Divider ↑ | Mean ↑ |
|---|---|---|---|---|---|---|---|
| OFT (Roddick et al., 2018) | 74.0 | 35.3 | 45.9 | 27.5 | 35.9 | 33.9 | 42.1 |
| LSS (Philion & Fidler, 2020) | 75.4 | 38.8 | 46.3 | 30.3 | 39.1 | 36.5 | 44.4 |
| CVT (Zhou & Krähenbühl, 2022) | 74.3 | 36.8 | 39.9 | 25.8 | 35.0 | 29.4 | 40.2 |
| M$^2$BEV (Xie et al.) | 77.2 | – | – | – | – | 40.5 | – |
| BEVFusion (Liu et al., 2023) | **81.7** | 54.8 | 58.4 | 47.4 | 50.7 | 46.4 | 56.6 |
| PolarFormer (Jiang et al., 2023) | 81.0 | 48.9 | 55.8 | - | 52.6 | 42.2 | - |
| **MosaicVT** | **81.7** | **55.4** | **58.8** | **48.4** | 52.8 | **46.8** | **57.3** |

## 4 EXPERIMENTS

**Setup** We evaluate MosaicVT on the nuScenes (Caesar et al., 2020) validation set for both 3D object detection and BEV segmentation using a camera-only, single-frame setting. All models are trained on the nuScenes training set. For detection, MosaicVT is integrated into the BEVDet (Huang et al., 2021) framework by replacing the view transformation module. For segmentation, MosaicVT is applied within the BEVFusion (Liu et al., 2023) pipeline using the same encoder, decoder, and training configuration. Models used for comparison vary in architecture but operate under similar conditions in terms of backbone capacity and input modality. All analysis and ablation studies are conducted using the BEVDet-based ResNet-50 detection setup with NVIDIA 3090 GPUs. Full training details are provided in Appendix.

### 4.1 MAIN RESULTS

**3D Object Detection** We follow the ResNet-50 experimental protocol established in Width-Former (Yang et al., 2024), which evaluates different view transformation modules under a unified BEVDet setup. MosaicVT is compared with both geometry-based methods (e.g., IPM, LSS, MatrixVT) and transformer-based models (e.g., BEVFormer, WidthFormer), all using the same backbone and training configuration. As shown in Table 1 (upper block), MosaicVT achieves the best performance across all metrics, surpassing prior methods in both detection accuracy and localization precision, without requiring global attention.

To assess scalability, We additionally report results with a ResNet-101 backbone and higher-resolution inputs, alongside other camera-only models that do not incorporate temporal fusion or additional supervision. As shown in Table 1 (lower block), MosaicVT consistently outperforms these models, demonstrating strong generalization across backbone scales and input settings.

**BEV Map Segmentation** For semantic segmentation, MosaicVT is evaluated within the camera-only BEVFusion (Liu et al., 2023) pipeline, using the same encoder and decoder while replacing only the VT module. We compare with recent BEV-based segmentation models that report results on nuScenes using similar backbone capacity and without access to LiDAR or temporal input. As shown in Table 2, MosaicVT achieves the highest IoU across all semantic classes, demonstrating strong generalization and spatial consistency in dense prediction tasks.

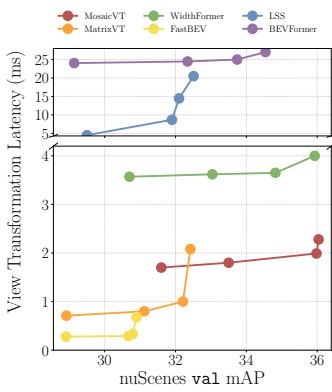

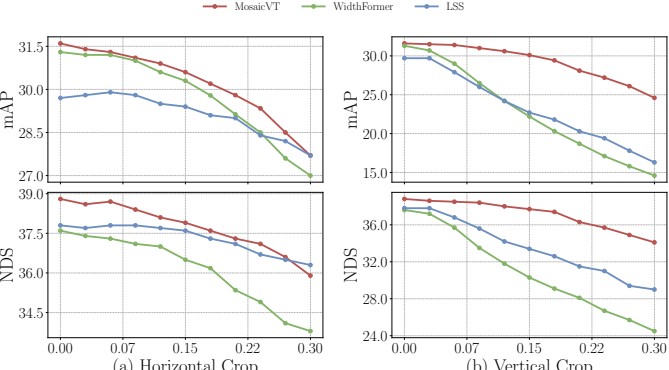

Figure 6: **Latency vs. mAP.** View transformation latency and mAP on nuScenes `val` set at input sizes (256,704), (384,1056), (512,1408) with representation dimension 64, and (512,1408) with dimension 128.

Figure 7: **Robustness to varying camera configurations.** Evaluation of model performance under simulated camera configuration changes. MosaicVT is compared with WidthFormer and LSS across two degradation scenarios: (a) Horizontal cropping simulates yaw shift and reduced horizontal FoV. (b) Vertical cropping simulates pitch change, mounting height, or vertical FoV variation.

**Temporal Extension** We evaluate the effectiveness of MosaicVT when extended to a multi-frame setting using BEVDet4D with 2-frame inputs. As shown in Table 3, MosaicVT achieves the best performance among all compared methods, attaining the highest mAP and NDS on the nuScenes validation set. These results demonstrate that MosaicVT not only performs well in single-frame scenarios but also generalizes effectively to temporal fusion, maintaining its advantage across view transformation methods.

Table 3: **Multi-frame results.**

| Method | mAP ↑ | NDS ↑ |
|---|---|---|
| IPM | 27.1 | 41.0 |
| LSS | 32.8 | 45.7 |
| MatrixVT | 32.4 | 45.8 |
| FastBEV | 30.8 | 42.4 |
| BEVFormer | 31.1 | 41.1 |
| WidthFormer | 34.0 | 46.3 |
| **MosaicVT** | **34.2** | **47.3** |

## 4.2 ANALYSIS

**Latency** We analyze the efficiency of MosaicVT in terms of view transformation latency. As shown in Figure 6, MosaicVT outperforms transformer-based methods like BEVFormer and WidthFormer in both mAP and latency across input sizes. Compared to lightweight models such as FastBEV and MatrixVT, it offers significantly stronger accuracy while maintaining comparable latency.

Table 4 details the source of this efficiency. Width-Former reduces cost via height-wise pooling and refinement, whereas MosaicVT performs camera-wise cross-attention in parallel, followed by lightweight BEV transformation and mosaic fusion. This parallel design eliminates heavy preprocessing and yields substantially lower latency.

Table 4: **Component-wise latency comparison.** Latency (ms) of WidthFormer and MosaicVT by major components. WidthFormer uses global cross-attention on pooled features, while MosaicVT adopts parallel camera-wise cross-attention with lightweight BEV transformation and mosaic fusion.

| **WidthFormer** | | **MosaicVT** | |
|---|---|---|---|
| Component | Latency | Component | Latency |
| Pool & Refine | 2.14 | - | - |
| Transformer | 1.43 | Transformer | 0.93 |
| - | - | BEV Trans. | 0.41 |
| - | - | Mosaic Fusion | 0.36 |
| Total | 3.57 | Total | **1.70** |

**Camera Configuration** We test robustness to camera configuration changes using test-time cropping, which simulates yaw, pitch, and FoV variations without additional data. As shown in Figure 7, MosaicVT consistently outperforms WidthFormer across degradation levels and achieves comparable or higher accuracy than LSS, which is naturally robust due to direct 3D lifting.

To further evaluate robustness, we introduced random perturbations to camera extrinsics ($\leq 1°$ rotation, $\leq 2$ cm translation), reflecting typical calibration errors (Qin et al., 2024). As summarized in Table 5, MosaicVT sustains SOTA-level performance under these conditions, confirming resilience to realistic deviations in practice.

Table 5: **Robustness to camera extrinsic calibration errors.** [†] indicates evaluation with added random perturbation noise ($\leq 1°$ rotation, $\leq 2$ cm translation).

| Method | mAP ↑ | NDS ↑ |
|---|---|---|
| LSS | 29.5 | 37.1 |
| WidthFormer | 30.7 | 37.3 |
| **MosaicVT[†]** | **30.8** | **37.7** |
| **MosaicVT** | **31.6** | **39.3** |

Table 6: **Positional embedding.** Ablation with camera-centric polar $(\theta^v, r^v, h)$ as baseline, showing drops from removing h, Cartesian, or ego-centric.

| Positional Encoding | $\Delta$mAP | $\Delta$NDS |
|---|---|---|
| $\theta^{ego}, r^{ego}, h$ | -0.8 | -1.6 |
| $x^v, y^v, h$ | -0.6 | -0.3 |
| $\theta^v, r^v$ | -0.1 | -0.7 |

Table 7: **BEV transformation.** Ablation on condition types for BEV transformation: none, ego BEV position, and camera extrinsics $(\text{FoV}_{ego}, \mathbf{z}_{ego}, \mathbf{o}_{ego})$.

| Transform Condition | mAP ↑ | NDS ↑ |
|---|---|---|
| - | 30.2 | 37.6 |
| global BEV position | 30.3 | 37.5 |
| $\text{FoV}_{ego}, \mathbf{z}_{ego}, \mathbf{o}_{ego}$ | **31.6** | **39.3** |

Table 8: **BEV representation aggregation.** Evaluation of fusion strategies for BEV representation on all objects and those visible in multiple cameras.

| View | Agg. | $\text{mAP}_{all}$ ↑ | $\text{mAP}_{overlap}$ ↑ |
|---|---|---|---|
| Multi | - | 30.7 | 31.3 |
| Single | Max | 31.4 | 31.4 |
| Single | Avg | **31.6** | **31.6** |

Table 9: **BEV resolution.** Evaluation of combinations of ego and camera polar BEV resolutions in single-view transformation.

| Ego | Camera | mAP ↑ | NDS ↑ |
|---|---|---|---|
| 128×128 | 16×44 | **31.6** | **39.3** |
| 128×128 | 8×22 | 0.4 | 2.7 |
| 64×64 | 16×44 | 25.6 | 33.0 |

## 4.3 ABLATION STUDIES

**View-Agnostic Positional Embedding**  We compare positional encoding strategies with camera-centric polar $(\theta^v, r^v, h)$ as the baseline. As shown in Table 6, dropping height $h$ degrades performance, highlighting the role of vertical cues. Replacing polar with Cartesian also degrades results, while camera-centric encoding outperforms ego-centric, indicating reduced spatial ambiguity.

**BEV Transformation**  Table 7 compares strategies for aligning camera-centric BEV representations to the global space. Removing the transformation leads to sharp performance drops, indicating the need to align spatial cues in the representation. Conditioning on camera extrinsics outperforms using only global positions, highlighting the effectiveness of incorporating scale, rotation, and translation.

**Mosaic Fusion**  We evaluate fusion strategies for overlapping BEV representations. Table 8 compares multi-view and single-view approaches using max or average pooling. Evaluation uses nuScenes `val` set, comparing results on all objects and those visible in multiple cameras. Average pooling slightly outperforms max pooling, with both yielding similar performance on all and overlapping objects. While the multi-view baseline benefits from correspondence in overlapping regions, MosaicVT outperforms it across the board, indicating stronger alignment without explicit cross-view attention.

**BEV Resolution**  Table 9 evaluates the impact of BEV grid resolutions in single-view transformation. Performance degrades sharply when the resolution of the camera polar BEV is reduced, even with a high-resolution ego BEV. In contrast, lowering only the ego BEV resolution causes a smaller drop, indicating that the camera polar BEV plays a more critical role in preserving spatial detail.

## 5 CONCLUSION

We introduce MosaicVT, a framework for multi-camera BEV representation learning that unifies camera-centric transformations with structured geometric alignment and selective fusion. By independently transforming each view and projecting it into a global BEV frame , MosaicVT reduces cross-view redundancy and enhances spatial coherence. Experiments on nuScenes show that MosaicVT achieves competitive or superior performance across multiple perception tasks with notable efficiency gains over transformer-based methods. It further exhibits strong robustness under variations in camera configuration and calibration noise, underscoring its adaptability to real-world scenarios.

**Limitations**  MosaicVT assumes accurate extrinsics and a relatively rigid mounting setup. While our experiments show robustness to realistic calibration noise, accumulated or larger errors, as well as the need for recalibration in dynamic settings, remain important direction for future work.

**Broader Impacts**  Our method contributes to efficient and robust 3D perception in safety-critical applications such as autonomous driving. By reducing computation while maintaining high spatial fidelity, MosaicVT opens up deployment opportunities in resource-constrained environments.

## REPRODUCIBILITY STATEMENT

We have made extensive efforts to ensure the reproducibility of our work. All architectural details of the proposed Mosaic View Transformation (MosaicVT) framework, including the design of the view-agnostic positional embedding, single-view transformation, and mosaic fusion strategy, are provided in the main text (Sections 3) with precise mathematical formulations and diagrams. Implementation details such as model resolutions, feature extraction, training schedules, data augmentations, and optimizer configurations are described in the Appendix A. We conduct experiments on the publicly available nuScenes dataset (Caesar et al., 2020), and the preprocessing steps and evaluation protocols strictly follow established benchmarks (Section 4). Ablation studies isolating key components (Table 6, 7, 8, and 9) and analyses of efficiency and robustness (Section 4.2) further support the transparency of our findings. Together, these descriptions provide sufficient details for reproducing our results without requiring additional proprietary resources.

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

## A ADDITIONAL TRAINING DETAILS

For 3D object detection, we follow the training setup of WidthFormer Yang et al. (2024). We use ResNet-50 or ResNet-101 backbones with input resolutions of $256 \times 704$ and $512 \times 1408$, respectively. Image features are extracted at $1/16$ resolution and projected to a $128 \times 128$ BEV grid covering a $102.4\,\text{m} \times 102.4\,\text{m}$ area. The model is trained for $24$ epochs using the AdamW optimizer with an initial learning rate of 2e-4, step decay at epochs 16 and 22, and weight decay of 1e-2. The total batch size is 32. Data augmentations include horizontal flipping, global scaling, and mild BEV-space rotation.

For BEV semantic segmentation, we follow the configuration of BEVFusion Liu et al. (2023). The BEV grid is set to $256 \times 256$, covering the same $102.4\,\text{m} \times 102.4\,\text{m}$ area. We use the original segmentation decoder and train for 20 epochs using the AdamW optimizer with an initial learning rate of 1e-4, weight decay of 1e-2, a cyclic learning rate schedule, and linear warmup. The batch size is 32. BEV-space augmentations include random rotation, scaling, and translation. Cross-entropy loss is used for supervision.

## B THE USE OF LARGE LANGUAGE MODELS (LLMs)

During the preparation of this paper, large language models were employed in a restricted and supervised manner. Their role was restricted to English proofreading and grammar correction. They were not employed to draft complete sentences or passages, nor to generate novel methods or results.

