# OpenReview forum: "Efficient Multi-View 3D Representation via Fusion of View-Agnostic Transformations"
_ICLR.cc/2026/Conference — Submitted to ICLR 2026_

### Official Review · Reviewer_Rq2n · 2025-10-26

**Soundness:** 3
**Presentation:** 3
**Contribution:** 3
**Rating:** 6
**Confidence:** 3

**Summary:**

The paper presents a novel framework for efficient multi-camera BEV representation in autonomous driving.
The method addresses redundancy in transformer-based view transformation by processing each camera view independently in a camera-centric polar coordinate system.
Key innovations include View-Agnostic Positional Embedding (VAPE) for consistent 3D localization and a modular transformation-fusion pipeline that avoids cross-view attention.
Experiments on nuScenes show that MosaicVT outperforms existing methods in 3D detection and BEV segmentation while reducing latency.
It also demonstrates robustness to camera perturbations and configuration changes.

**Strengths:**

1. Computational Efficiency and Reduced Redundancy. MosaicVT eliminates the computationally expensive global cross-view attention used in transformer-based methods.
2. Novel and Robust Formulation. The introduction of the camera-centric polar coordinate system and View-Agnostic Positional Embedding (VAPE) effectively resolves spatial ambiguity inherent in image-to-BEV transformation.
3. The paper is well-written and easy to follow.

**Weaknesses:**

1. The method is based on the assumption that cross-view interaction is not important for global BEV. It might be true for nuScenes with little FoV overlap between cameras. However, the paper does not take cases into consideration where there might be considerable FoV overlap between surround cameras.
2. The paper does not compare against recent state-of-the-art methods for object detection and BEV segmentation.

**Questions:**

1. As in the weakness section, could you please discuss the validity of the assumption that cross-view attention is negligible?
2. Why do you only compare with methods that is more than one year from now?

---

> ### Author Response · Authors · 2025-11-22
> **Official Comment by Authors**
>
> Thank you for taking the time to thoroughly review our manuscript.
>
> In response to your detailed feedback, we have gone to great lengths to address and accommodate every single one of your comments.
>
> We would greatly appreciate it if you could review our responses to your comments.
>
> Sincere thanks in advance for your time and efforts.
>
> ### **Weakness 1 & Question 1. Cross-view interaction**
>
> We agree with the reviewer that nuScenes has limited FoV overlap, and this naturally reduces the contribution of cross-view interaction. We acknowledge this dataset property. However, our intention is not to claim that cross-view cues are unimportant in general; rather, we show that global cross-view attention, as used in transformer-based VT, provides limited benefit while incurring substantial computational cost. As shown in Fig. 1(a), BEV queries overwhelmingly attend to the camera that directly observes each region, even in partially overlapping areas.
>
> MosaicVT is designed to first perform per-camera view transformation and then combine these camera-centric BEV representations through an efficient geometry-aware fusion stage. While our fusion uses simple averaging, it still merges multi-camera information effectively without the computation cost of global cross-view attention.
>
> To evaluate whether this simple strategy actually leverages information from overlapping camera regions, we report $\text{mAP}_\text{overlap}$ in Table 8. This metric measures detection performance specifically on objects that are visible in more than one camera, which directly reflects how well a model aggregates signals from multiple viewpoints. The results show that average fusion outperforms max fusion, indicating that combining multiple camera features is beneficial. In addition, average fusion also achieves higher accuracy than the multi-view attention baseline. These findings suggest that the proposed aggregation method, although simple, effectively integrates complementary cues from overlapping views and performs reliably within the MosaicVT framework.
>
> ### **Weakness 2 & Question 2. Comparisons with recent SOTAs**
>
> Our work focuses on the view transformation (VT) stage, which converts multi-view image features into BEV representations. Therefore, our comparison targets methods that propose new VT mechanisms. Many recent approaches improve other components, such as detector heads, dense BEV encoders, or temporal fusion modules, and thus cannot be included in a VT-level comparison.
>
> However, a recently proposed VT method, GeoBEV[1], does introduce a new BEV-based VT formulation and achieves stronger accuracy than other recent camera-only 3D detectors in full detection pipelines. This makes GeoBEV the strongest existing BEV-based detector and the most relevant baseline for assessing VT quality. We therefore include GeoBEV alongside LSS, MatrixVT, FastBEV, BEVFormer, and WidthFormer. All VT methods are evaluated under the same BEVDet + ResNet-50 configuration to ensure fairness. As shown in Table A, MosaicVT achieves the strongest performance among comparable VT modules. We will incorporate the newly added VT-only comparisons, including the GeoBEV experiment, into the revised manuscript.
>
> **Table A.** 3D object detection.
>
> | Method | mAP | NDS |
> | --- | --- | --- |
> | WidthFormer | 30.7 | 37.3 |
> | GeoBEV | 31.0 | 39.1 |
> | MosaicVT (Ours) | **31.6** | **39.3** |
>
> ### **Reference**
>
> [1] Zhang, Jinqing, et al. "Geobev: Learning geometric bev representation for multi-view 3d object detection." Proceedings of the AAAI Conference on Artificial Intelligence. Vol. 39. No. 9. 2025.

---

### Official Review · Reviewer_hBDg · 2025-10-27

**Soundness:** 3
**Presentation:** 3
**Contribution:** 3
**Rating:** 6
**Confidence:** 3

**Summary:**

This paper proposes a modular framework, MosaicVT, that generates a unified BEV representation by independently transforming multiple camera viewpoints. MosaicVT uses a central camera polar-coordinate system to address directional ambiguity and multi-view redundancy effectively. Through novel view-independent position embeddings, MosaicVT can generalize to different camera configurations without retraining. The generated camera-centered BEV representations are aligned and fused using a geometry-aware interpolation strategy, thereby significantly reducing computational overhead while maintaining high accuracy and robustness. Experimental results show that MosaicVT achieves state-of-the-art performance on 3D object detection and BEV semantic segmentation tasks on the nuScenes benchmark, while significantly reducing latency and performing robustly across different camera configurations.

**Strengths:**

Different from previous methods that uniformly process the global view, we propose a method that processes each camera view independently and then aggregates them into a unified BEV space, effectively reducing cross-view interference and spatial ambiguity while avoiding the computational overhead of the global attention mechanism.
The method introduction is clear and logical. The authors explain the core idea through detailed formula derivation and provide complete mathematical proofs for key details.
In the experimental part, the effectiveness of introducing camera-centered polar coordinates as position embedding is effectively proved by detailed comparison experiments and ablation experiments

**Weaknesses:**

1. Using polar coordinates for position encoding instead of Cartesian coordinates is actually a relatively common idea, which may not be very innovative.
2. When feature conflicts exist between different views, simple averaging and 2D convolution lack a dynamic arbitration mechanism to intelligently decide which view should be preferentially adopted, which may perform poorly when dealing with complex occlusions and view conflicts compared with more advanced fusion modules, such as attention-based fusion modules.

**Questions:**

Compared with the previous unified global-view processing method, is the weighted-average fusion method and 2D convolution adopted in this paper too simple?
Although this lightweight design is efficient, is its ability to aggregate information sufficient?
Are there more advanced fusion methods (e.g., cross-view attention) that can handle information interaction between views more robustly?

---

> ### Author Response · Authors · 2025-11-22
> **Official Comment by Authors**
>
> Thank you for taking the time to thoroughly review our manuscript.
>
> In response to your detailed feedback, we have gone to great lengths to address and accommodate every single one of your comments.
>
> We would greatly appreciate it if you could review our responses to your comments.
>
> Sincere thanks in advance for your time and efforts.
>
> ### **Weakness 1. Polar coordinate**
>
> We agree that polar-coordinate formulations have appeared in prior works such as PolarFormer[1] and GeoBEV[2]. However, MosaicVT’s contribution does not lie in using polar coordinates themselves. Instead, our novelty is in enabling a shared, lightweight, and camera-agnostic learnable lifting module that resolves redundant cross-view attention and 2D→3D ambiguity.
>
> To achieve this, MosaicVT performs per-camera lifting with a single shared network, which is made possible by expressing each pixel in normalized camera-centric geometry. Polar coordinates are therefore used not as a standalone innovation, but as a necessary representation to eliminate directional ambiguity and to support a unified, view-agnostic lifting process. Our contribution is the integration of this geometry with camera-agnostic learnable lifting, not the coordinate choice itself.
>
> ### **Weakness 2 & Question. Simple aggregation**
>
> We appreciate the reviewer’s question regarding whether the cross-view fusion in MosaicVT is overly simple. Our design performs per-camera view transformation first, and the fusion stage then provides an efficient way to combine information from multiple cameras. Although averaging fusion are lightweight, they still allow different camera features to be merged without the computational overhead of global cross-view attention.
>
> To evaluate whether this simple strategy actually leverages information from overlapping camera regions, we report $\text{mAP}_\text{overlap}$ in Table 8. This metric measures detection performance specifically on objects that are visible in more than one camera, which directly reflects how well a model aggregates signals from multiple viewpoints. The results show that average fusion outperforms max fusion, indicating that combining multiple camera features is beneficial. In addition, average fusion also achieves higher accuracy than the multi-view attention baseline. These findings suggest that the proposed aggregation method, although simple, effectively integrates complementary cues from overlapping views and performs reliably within the MosaicVT framework.
>
> We also appreciate the reviewer’s suggestion that more advanced cross-view interactions, such as attention-based fusion or dynamic weighting, could further improve performance. While this paper focuses on an efficient and modular design, exploring richer fusion mechanisms or hybrid multi-view architectures is a promising direction for future research.
>
> ### **Reference**
>
> [1] Jiang, Yanqin, et al. "Polarformer: Multi-camera 3d object detection with polar transformer." Proceedings of the AAAI conference on Artificial Intelligence. Vol. 37. No. 1. 2023.
>
> [2] Zhang, Jinqing, et al. "Geobev: Learning geometric bev representation for multi-view 3d object detection." Proceedings of the AAAI Conference on Artificial Intelligence. Vol. 39. No. 9. 2025.

---

> > ### Comment · Reviewer_hBDg · 2025-11-26
> >
> > Based on the rebuttal, the authors have addressed my concerns, and i will keep my positive rating.

---

> > > ### Author Response · Authors · 2025-11-26
> > > **Official Comment by Authors**
> > >
> > > Thank you very much for your thoughtful evaluation and for taking the time to review our rebuttal. We sincerely appreciate your positive assessment and your constructive feedback throughout the process.

---

### Official Review · Reviewer_bMFc · 2025-10-28

**Soundness:** 3
**Presentation:** 2
**Contribution:** 3
**Rating:** 4
**Confidence:** 4

**Summary:**

The paper proposes MosaicVT, a BEV view-transformation module that (i) transforms each camera independently in a camera-centric polar frame (angles θ, radial distance r, and height h), (ii) makes the transformation view-agnostic via a positional encoding that uses relative geometric cues so one shared module can generalize across heterogeneous camera setups, and (iii) aligns & fuses the per-camera BEV “tiles” into a global BEV using geometry-aware interpolation—avoiding global cross-view attention altogether

**Strengths:**

The paper pinpoints two concrete issues—redundant cross-view referencing and ego-centric ray ambiguity and ties each to a design choice (per-camera VT + camera-centric polar coordinates), giving a clean problem, method story.

MosaicVT swaps only the view transformation stage inside standard pipelines (e.g., BEVDet for detection, BEVFusion for BEV segmentation), leaving heads/decoders unchanged—useful for real systems.

**Weaknesses:**

1. Motivation:
  a. The paper says prior view-transform “uses global cross-view attention,” but BEVFormer uses deformable (sparse) cross-attention from BEV queries, and LSS lifts frustums then splats to BEV—no global attention. Please clarify.
  b. The “A core challenge of this design is ensuring that a single transformation module can operate across heterogeneous camera setups” claim lacks a baseline analysis: why can’t BEVFormer/LSS work across heterogeneous camera?

2. Performance:
  a. Temporal length/stride/cache aren’t specified, and some numbers seem below commonly reported BEVFormer-base ≈ 0.517 NDS / 0.416 mAP on nuScenes—please detail the exact setting and explain the gap.
  b. Limited comparison coverage. Add more recent multi-view detectors (or justify exclusions) (e.g. Far3D, BEVNeXt, GeoBEV)

**Questions:**

1. In Table 9, why Camera $8\times 22$ results in poor performance?

---

> ### Author Response · Authors · 2025-11-22
> **Official Comment by Authors**
>
> Thank you for taking the time to thoroughly review our manuscript.
>
> In response to your detailed feedback, we have gone to great lengths to address and accommodate every single one of your comments.
>
> We would greatly appreciate it if you could review our responses to your comments.
>
> Sincere thanks in advance for your time and efforts.
>
> ### **Weakness 1. Motivation**
>
> Our motivation focuses specifically on learnable VT methods such as BEVFormer and PolarFormer, not explicit lifting methods like LSS. These attention-based approaches perform global multi-view interactions where each BEV query searches across all camera features. This design inherently depends on a fixed camera index and view layout, making the transformation brittle when camera order, FoV, or rig configuration changes.
>
> Although LSS does not use attention and is more robust under extrinsic perturbations, it projects features purely through depth estimation. Importantly, a 2D feature at a given pixel represents information captured from the specific viewing direction of that camera. It is not a direction-agnostic 3D descriptor. Therefore, even with accurate depth, simply relocating the feature to the corresponding 3D point cannot compensate for changes in camera orientation, because the feature itself was generated under a different directional context.
>
> MosaicVT addresses both limitations. Each camera is first transformed into a normalized polar BEV, which removes the dependency on camera-indexed global attention. A camera-aware BEV alignment step then incorporates each camera’s position, orientation, and field of view when fusing the per-camera BEV tiles into the global BEV. This design preserves the semantic benefits of learnable VT while ensuring robust, view-agnostic operation across heterogeneous camera setups.
>
> Thank you for highlighting this point. We will revise the manuscript to more clearly explain this issue.
>
> ### **Weakness 2. Performance**
>
> The BEVFormer results in our paper follow the WidthFormer evaluation setting. Concretely, we use the BEVDet framework with a ResNet-50 backbone and 256×704 input resolution, along with the 6-layer BEVFormer implementation provided by BEVDet. This setting is the standard VT-only protocol used in WidthFormer, MatrixVT, and LSS, and is designed to isolate the impact of the view-transformation module while keeping the rest of the detector fixed.
>
> Our comparisons therefore focus on methods that introduce new VT mechanisms. Far3D[1] and BEVNeXt[2] modify the overall 3D detection architecture (e.g., adaptive 3D queries, depth denoising, temporal BEV encoders) and do not propose a standalone VT module, so including them would not provide a controlled VT-level comparison.
>
> GeoBEV[3], in contrast, is a VT-specific method, and we evaluate it under the same BEVDet + ResNet-50 configuration to ensure a fair and controlled comparison. As shown in Table A, MosaicVT achieves higher mAP and NDS compared to recent VT methods, demonstrating the effectiveness of our approach. We will incorporate the newly added VT-only comparisons, including the GeoBEV experiment, into the revised manuscript.
>
> **Table A.** 3D object detection.
>
> | Method | mAP | NDS |
> | --- | --- | --- |
> | WidthFormer | 30.7 | 37.3 |
> | GeoBEV | 31.0 | 39.1 |
> | MosaicVT (Ours) | **31.6** | **39.3** |
>
> ### **Question. 8x22 resolution performance**
>
> The large drop at 8x22 occurs because the camera polar grid is the first and primary discretization of the image features when they are mapped into BEV space. This grid determines how much geometric detail from each view can be preserved before the per-camera BEV tiles are constructed. When the angular and radial resolution is too coarse, important directional and distance variations are heavily quantized, and the resulting tiles lose meaningful spatial structure.
>
> Since the global BEV is assembled directly from these per-camera tiles, the initial quantization errors propagate and become amplified during alignment, producing severe misalignment across views. In contrast, lowering the resolution of the global BEV has only a small impact (Table 9), which confirms that the polar grid is the true bottleneck that determines geometric fidelity.
>
> ### **Reference**
>
> [1] Jiang, Xiaohui, et al. "Far3d: Expanding the horizon for surround-view 3d object detection." Proceedings of the AAAI conference on artificial intelligence. Vol. 38. No. 3. 2024.
>
> [2] Li, Zhenxin, et al. "Bevnext: Reviving dense bev frameworks for 3d object detection." Proceedings of the IEEE/CVF conference on computer vision and pattern recognition. 2024.
>
> [3] Zhang, Jinqing, et al. "Geobev: Learning geometric bev representation for multi-view 3d object detection." Proceedings of the AAAI Conference on Artificial Intelligence. Vol. 39. No. 9. 2025.

---

> > ### Comment · Reviewer_bMFc · 2025-11-26
> >
> > After reading the rebuttal, I appreciate the additional context, but my main concerns are largely unchanged.
> >
> > (1) **Motivation and the “global cross-view attention” claim.**
> > The rebuttal still does not really address my original point that BEVFormer does *not* use dense global cross-view attention. Its spatial cross-attention is based on deformable attention, where each BEV query only samples a small set of learned offsets in the multi-view features rather than attending densely to all pixels in all cameras. This design is already substantially more structured and sparse than what the paper describes as “global cross-view attention”. Moreover, it is unclear why the authors believe that “this design inherently depends on a fixed camera index and view layout, making the transformation brittle when camera order, FoV, or rig configuration changes”; the rebuttal does not explain what specific aspect of BEVFormer’s deformable cross-attention leads to this claimed dependency. As it stands, the motivation somewhat overstates the limitations of prior VT methods, while the actual heterogeneity issues are not quantitatively demonstrated.
> >
> > (2) **Choice of baselines and experimental scope.**
> > I understand the authors’ intent to focus on “VT-only” modules under a fixed BEVDet+ResNet-50 protocol. However, my original concern was about the *breadth and practical relevance* of the comparisons. Modern camera-based 3D detectors such as Far3D and BEVNeXt integrate stronger architectures and represent the current state of the field. Excluding them on the grounds that they are “not standalone VT modules” significantly limits the scope and practical significance of the empirical validation: from an application perspective, one would like to know whether proposed method still brings gains when plugged into stronger, state-of-the-art multi-view detectors, not only within an older baseline.
> >
> > In summary, while the response clarifies some design choices, I still find the motivation somewhat overstated and the experimental evidence constrained to a narrow setting, which weakens the overall impact and generality of the work.

---

> ### Author Response · Authors · 2025-11-26
> **Official Comment by Authors**
>
> We thank the reviewer for the thoughtful follow-up and for highlighting the points that required clearer explanation. We address these concerns in detail below.
>
> ### **1. Motivation and BEVFormer clarity**
>
> We agree with the reviewer that BEVFormer avoids redundant global attention through deformable sampling. Our goal was not to claim dense attention, but to highlight limitations that deformable sampling does not address.
>
> Although deformable attention reduces redundancy by sparsely sampling a small number of pixel locations, its query-dependent and irregular memory access pattern results in poor practical latency on GPU hardware (Fig. 6). More importantly, deformable attention samples 2D image features that are spatially aggregated rather than point-sampled. When the camera setup changes, such as variations in camera pose, field of view, the number of cameras, the 3D space that a BEV query corresponds to is still covered by some sampled features, but the resulting 3D position distribution of the sampled features changes. Since BEVFormer samples and uses these features without any explicit 3D positional information or camera-related cues, it has no mechanism or information to account for these changes.
>
> WidthFormer removes deformable attention to improve practical efficiency and injects pixel-level geometric positional encoding so that each feature retains its 3D position. However, an image feature is fundamentally a view-dependent observation of a 3D point: a particular direction and field of view determine how that point appears in the image. When the camera setup changes, even if the 3D position encoded by the positional embedding is the same, the corresponding image feature reflects a different view. WidthFormer does not encode camera-level extrinsics, so it assumes the feature is view-invariant and cannot compensate for these view-dependent changes, which leads to the degradation observed in Fig. 7.
>
> MosaicVT resolves these issues by transforming each camera into a normalized polar frame and applying a camera-aware BEV alignment that explicitly incorporates the camera orientation and field of view. We will clarify this explanation in the revision, and we appreciate the reviewer for pointing out the need for a more precise description.
>
> ### **2. Baseline scope and practical relevance**
>
> We appreciate the reviewer’s concern about the broader empirical scope. BEV view transformation provides general-purpose BEV features that support multiple tasks such as 3D detection and segmentation. At the same time, we agree that comparing against strong task-specific architectures is important for understanding practical relevance, even when those methods do not use BEV features.
>
> Far3D[1] and RayDN[2] follow DETR-style pipelines and do not rely on BEV features or a VT module. BEVNeXt[3] is BEV-based but keeps the original LSS VT module unchanged while improving temporal fusion and detection head. Because MosaicVT replaces only the VT stage, it can be directly integrated into BEVNeXt-style architectures and is expected to strengthen them by providing higher-quality BEV features.
>
> GeoBEV[4] is not only the only method among these that introduces a new BEV-based VT mechanism, but it also represents the strongest existing BEV-based detector. In full detection pipelines, it reports higher accuracy than Far3D, RayDN, and BEVNeXt, making it the most competitive and relevant baseline for evaluating VT quality. Under the same BEVDet  configuration, MosaicVT surpasses GeoBEV in both mAP and NDS, indicating that our VT module outperforms the strongest existing BEV-based approach under fair and controlled comparison.
>
> Due to our computational constraints, we were unable to run full SOTA detectors. Instead, we focus on a fair experimental design where all components other than the VT module are kept identical, and under this fair setting MosaicVT exceeds GeoBEV, which itself already outperforms Far3D, RayDN, and BEVNeXt. We will clarify this rationale and scope in the revision.
>
> ### **Reference**
>
> [1] Jiang, Xiaohui, et al. "Far3d: Expanding the horizon for surround-view 3d object detection." Proceedings of the AAAI conference on artificial intelligence. Vol. 38. No. 3. 2024.
>
> [2] Liu, Feng, et al. "Ray denoising: Depth-aware hard negative sampling for multi-view 3d object detection." European Conference on Computer Vision. Cham: Springer Nature Switzerland, 2024.
>
> [3] Li, Zhenxin, et al. "Bevnext: Reviving dense bev frameworks for 3d object detection." Proceedings of the IEEE/CVF conference on computer vision and pattern recognition. 2024.
>
> [4] Zhang, Jinqing, et al. "Geobev: Learning geometric bev representation for multi-view 3d object detection." Proceedings of the AAAI Conference on Artificial Intelligence. Vol. 39. No. 9. 2025.

---

### Official Review · Reviewer_5ZeX · 2025-10-31

**Soundness:** 3
**Presentation:** 3
**Contribution:** 2
**Rating:** 4
**Confidence:** 5

**Summary:**

This paper proposes MosaicVT, a modular framework for multi-camera BEV representation learning, aiming to address the computational inefficiency from cross-view redundancy and spatial ambiguity in ego-centric projections of existing attention-based methods. Extensive experiments on the nuScenes benchmark demonstrate that MosaicVT achieves competitive performance in 3D object detection and BEV semantic segmentation, with substantially reduced latency compared to transformer-based methods. It also exhibits strong robustness to variations in camera configuration and calibration noise.

**Strengths:**

1. MosaicVT processes each camera view independently using a camera-centric polar coordinate system, which avoids the unnecessary global cross-view attention in transformer-based methods.
2. The proposed VAPE embeds image features using relative geometric cues, abstracting away camera-specific parameters. This enables a single transformation module to adapt to diverse camera setups without retraining.
3. Experiments on simulated camera configuration changes and real-world calibration noise  show that MosaicVT outperforms WidthFormer and LSS in robustness.

**Weaknesses:**

1. The method of converting image features into Polar BEV and then obtaining BEV features through sampling is somewhat similar to the RC-Sample proposed by GeoBEV[1]. The advantages of MosaicBEV need to be further demonstrated.
2. MosaicBEV has not been compared with current SOTA methods, such as RayDN[2], BEVNext[2] and so on.
3. In Table 4, MosaicBEV is only compared with WidthFormer in terms of efficiency. The efficiency comparison with LSS-based methods should also be added.

[1] Zhang, Jinqing, et al. "Geobev: Learning geometric bev representation for multi-view 3d object detection." Proceedings of the AAAI Conference on Artificial Intelligence. Vol. 39. No. 9. 2025.\
[2] Liu, Feng, et al. "Ray denoising: Depth-aware hard negative sampling for multi-view 3d object detection." European Conference on Computer Vision. Cham: Springer Nature Switzerland, 2024.\
[3] Li, Zhenxin, et al. "Bevnext: Reviving dense bev frameworks for 3d object detection." Proceedings of the IEEE/CVF conference on computer vision and pattern recognition. 2024.

**Questions:**

4. DFA3D also uses depth distribution to avoid the problem of depth uncertainty. What advantages does MosaicBEV have compared with DFA3D?
5. In Figure 7, why is the impact of vertical crop on LSS greater than that of horizontal crop?

---

> ### Author Response · Authors · 2025-11-22
> **Official Comment by Authors**
>
> Thank you for taking the time to thoroughly review our manuscript.
>
> In response to your detailed feedback, we have gone to great lengths to address and accommodate every single one of your comments.
>
> We would greatly appreciate it if you could review our responses to your comments.
>
> Sincere thanks in advance for your time and efforts.
>
> ### **Weakness 1. Comparison to GeoBEV**
>
> Thank you for highlighting the relation to GeoBEV[1], which indeed serves as an important baseline. We acknowledge that both GeoBEV and MosaicVT utilize a polar/radial representation followed by Cartesian sampling. However, GeoBEV’s RC-Sampling uses a predefined, non-learnable mapping to radial space, and its splatting to the Cartesian BEV does not account for camera geometry or directional differences, simply aggregating radial maps in a fixed manner. In contrast, MosaicVT provides a learnable, camera-agnostic transformation framework: (i) image features are first projected into camera-centric polar BEV using a shared network, (ii) a pose-conditioned learnable transformation aligns these per-camera BEVs to the global frame, and (iii) mosaic fusion integrates all views in a geometry-aware manner. This yields more reliable multi-view alignment and stronger adaptability across camera configurations, reflected in MosaicVT’s higher accuracy under identical BEVDet settings (31.6 vs. 31.0 mAP, Table A). We will incorporate a clearer discussion of these differences and explicitly compare MosaicVT with GeoBEV in the revised manuscript.
>
> ### **Weakness 2. Comparisons with recent SOTAs**
>
> Our work focuses specifically on the view transformation (VT) stage, that is, how multi-view image features are converted into BEV representations. For this reason, our comparisons target methods that also propose new VT mechanisms. RayDN[2] and BEVNeXt[3] do not introduce new VT modules. RayDN presents a depth-aware negative sampling strategy for DETR-based detectors, and BEVNeXt redesigns dense BEV encoders and detection heads. Because these methods improve components outside the VT stage, they are not included in our VT-level comparison.
>
> GeoBEV, in contrast, introduces a new BEV-based VT mechanism and achieves higher accuracy than RayDN and BEVNeXt in full detection pipelines. As a result, GeoBEV represents the strongest existing BEV-based detector and is the most relevant baseline for evaluating VT quality. As shown in Table A, under the same BEVDet + ResNet-50 configuration, MosaicVT surpasses GeoBEV in both mAP and NDS, indicating that our VT module outperforms the strongest previous BEV-based approach under fair and controlled comparison.
>
> **Table A.** 3D object detection.
>
> | Method | mAP | NDS |
> | --- | --- | --- |
> | WidthFormer | 30.7 | 37.3 |
> | GeoBEV | 31.0 | 39.1 |
> | MosaicVT (Ours) | **31.6** | **39.3** |
>
> ### **Weakness 3. Efficiency comparison with LSS**
>
> In Table 4 we report a component-wise latency breakdown for MosaicVT and WidthFormer, the previous SOTA transformer-based VT, because both methods decompose naturally into comparable stages such as attention, BEV transformation, and fusion. LSS-style lift–splat VTs do not have such corresponding components, since their operations are executed as a single projection pipeline. Therefore, LSS is compared only at the overall VT latency level (Figure 6), while the component-wise breakdown is used to highlight where MosaicVT achieves efficiency gains over transformer-based VT methods.

---

> ### Author Response · Authors · 2025-11-22
> **Official Comment by Authors**
>
> ### **Question 1. Comparison to DFA3D**
>
> DFA3D[4] reduces depth ambiguity by allowing BEV queries to sample from a global 3D volume through depth-enhanced deformable attention. MosaicVT differs fundamentally: it performs view transformation in a camera-centric polar space, where VAPE provides the missing per-ray 3D cues via r and h. This avoids operating on a full 3D volume and directly resolves ray ambiguity specific to each camera. As a result, MosaicVT achieves camera-agnostic behavior with much lower computational cost, while maintaining competitive accuracy. These properties constitute clear advantages over DFA3D.
>
> ### **Question 2. Sensitivity to vertical cropping**
>
> LSS relies on predicting a dense depth distribution for each pixel and lifting image features along the corresponding projection rays. Vertical cropping removes upper and lower image regions that contain critical geometric cues for depth estimation, such as the horizon, road boundaries, and long-range context. Because the predicted depth distributions become less reliable when these cues disappear, the lift–splat process accumulates larger projection errors, which leads to a larger performance drop for LSS under vertical cropping.
>
> ### **Reference**
>
> [1] Zhang, Jinqing, et al. "Geobev: Learning geometric bev representation for multi-view 3d object detection." Proceedings of the AAAI Conference on Artificial Intelligence. Vol. 39. No. 9. 2025.
>
> [2] Liu, Feng, et al. "Ray denoising: Depth-aware hard negative sampling for multi-view 3d object detection." European Conference on Computer Vision. Cham: Springer Nature Switzerland, 2024.
>
> [3] Li, Zhenxin, et al. "Bevnext: Reviving dense bev frameworks for 3d object detection." Proceedings of the IEEE/CVF conference on computer vision and pattern recognition. 2024.
>
> [4] Li, Hongyang, et al. "Dfa3d: 3d deformable attention for 2d-to-3d feature lifting." Proceedings of the IEEE/CVF International Conference on Computer Vision. 2023.

---

### Author Response · Authors · 2025-12-03
**Summary of Contributions and Discussion**

Dear Area Chair,

We thank the Area Chair and all reviewers for their careful evaluation of our submission. To assist the final decision process, we provide the following summary of the reviewers’ positive assessments and how our rebuttal addressed their key concerns.

---

## **Contribution Summary**

Across all reviews, several strengths of MosaicVT were consistently emphasized.

**Methodological Soundness and Clarity**

All reviewers noted that our modular VT pipeline, which includes per-camera lifting in normalized polar space, view-agnostic positional embedding (VAPE), and geometry-aware alignment, is conceptually well-motivated and clearly presented. Reviewers also highlighted that the paper provides a clean problem–method narrative and is easy to follow.

**Efficiency and Practicality**

Reviewers 5ZeX, hBDg, and Rq2n highlighted that MosaicVT removes the need for costly cross-view attention while maintaining strong accuracy. Reviewers also recognized that the method operates as a modular VT component that can be directly substituted into BEVDet and BEVFusion without changes to downstream heads or decoders.

**Robustness**

Reviewers 5ZeX, hBDg, and Rq2n found the robustness experiments convincing. MosaicVT shows stable performance under camera pose shifts, calibration noise, and broader camera-configuration changes, outperforming both LSS and transformer-based VT modules.

---

## **Discussion Summary**

**Baseline Scope (5ZeX, bMFc and Rq2n)**

We clarified that recent detectors such as Far3D[1], RayDN[2], and BEVNeXt[3] do not introduce new view transformation mechanisms. Instead, these methods modify downstream detection heads or temporal modules, which makes them unsuitable for a controlled VT-only comparison. GeoBEV[4], in contrast, proposes a view transformation mechanism and reports higher accuracy than Far3D, RayDN, and BEVNeXt in full detection pipelines. For this reason, GeoBEV represents the strongest and most relevant baseline for our setting. Under identical BEVDet with ResNet-50 conditions, MosaicVT outperforms GeoBEV with 31.6 mAP compared to 31.0 mAP (Table A). We also noted that MosaicVT can be incorporated into BEVNeXt by replacing its LSS-based lifting, which suggests that MosaicVT could further enhance BEVNeXt-style architectures.

**Table A.** 3D object detection.

|Method | mAP | NDS |
|--- | --- | --- |
|WidthFormer| 30.7 | 37.3 |
|GeoBEV| 31.0|39.1|
|MosaicVT (Ours)|**31.6**|**39.3**|

**Clarifying BEVFormer Motivation (bMFc)**

We clarified that the intention was not to describe BEVFormer as using dense global attention. BEVFormer reduces redundancy through deformable sampling, yet this sampling introduces irregular memory access patterns and does not include camera-aware geometric cues. As a result, the sampled features shift unpredictably when the camera pose or field of view changes. WidthFormer improves efficiency by removing deformable sampling and adding pixel-level geometric positional embeddings, but the underlying image features remain view-dependent observations. Since camera-level extrinsics are not encoded, the model cannot compensate for changes in viewpoint. MosaicVT resolves these limitations by lifting each camera into a normalized polar representation and applying explicit camera-aware BEV alignment. As shown in Figure 7, MosaicVT maintains stable performance under camera perturbations, whereas WidthFormer degrades.

**Fusion and Cross-view Interaction (hBDg and Rq2n)**

Reviewers questioned whether the simplicity of our fusion module might limit its ability to integrate multi-camera cues. We clarified that MosaicVT does not assume cross-view information is unimportant. Instead, camera-aware alignment ensures that complementary information from different views is already placed in a consistent geometric frame. After this alignment, lightweight averaging is sufficient to combine features effectively. This is supported by the results in Table 8. Average fusion outperforms both max fusion and an attention-based multi-view baseline. The strong performance on the $\text{mAP}_\text{overlap}$ subset further shows that MosaicVT successfully integrates information from objects observed by multiple cameras.

---

### **Reference**

[1] Jiang, Xiaohui, et al. "Far3d: Expanding the horizon for surround-view 3d object detection." Proceedings of the AAAI conference on artificial intelligence. Vol. 38. No. 3. 2024.

[2] Liu, Feng, et al. "Ray denoising: Depth-aware hard negative sampling for multi-view 3d object detection." European Conference on Computer Vision. Cham: Springer Nature Switzerland, 2024.

[3] Li, Zhenxin, et al. "Bevnext: Reviving dense bev frameworks for 3d object detection." Proceedings of the IEEE/CVF conference on computer vision and pattern recognition. 2024.

[4] Zhang, Jinqing, et al. "Geobev: Learning geometric bev representation for multi-view 3d object detection." Proceedings of the AAAI Conference on Artificial Intelligence. Vol. 39. No. 9. 2025.

---

> ### Author Response · Authors · 2025-12-03
> **Final Remark**
>
> We also note that, due to the recent issue that affected the review system, the discussion phase ended before reviewers could provide final confirmations regarding our responses. Although we were unable to continue the dialogue, we believe that our rebuttal and the additional experiments we provided, including the GeoBEV comparison, the WidthFormer clarification, and the $\text{mAP}_\text{overlap}$ analysis, sufficiently address the concerns raised by Reviewers 5ZeX, bMFc, hBDg, and Rq2n. We respectfully hope that these clarifications, along with the reviewers' consistently positive evaluations of MosaicVT, will be fully considered in the final decision.
>
> Sincerely,
>
> The Authors

---

### Meta-Review · Area_Chair_Qgve · 2026-01-02

**Summary:**

The paper proposes MosaicVT, a BEV view-transformation module for multi-camera BEV representation learning. The main concerns are unclear motivation, missing SOTAs comparison, marginal improvement. A rebuttal is provided to partially addressed the concerns, but this paper needs significant revision for meeting the bar of ICLR.

**Reviewer Concerns:**

Concerns of Reviewer 5ZeX is not fully addressed.
Concerns of Reviewer bMFc is not addressed.
Concerns of Reviewer hBDg is addressed.
Concerns of Reviewer Rq2n is not fully addressed.

**Reviewer Scores:**

Reviewer 5ZeX would not change their score.
Reviewer bMFc would not change their score.
Reviewer hBDg would not change their score.
Reviewer Rq2n would not change their score.

---

### Decision · Program_Chairs · 2026-01-26

Reject